# The Ethical Foundations of Being Respected as a Citizen: A Triadic Theory Consisting of Autonomy, Boundary, and Equality

**DOI:** 10.3390/bs15040513

**Published:** 2025-04-11

**Authors:** Chuanjun Liu, Li Zhou

**Affiliations:** 1Department of Sociology and Psychology, School of Public Administration, Sichuan University, Chengdu 610065, China; 2024225010007@stu.scu.edu.cn; 2Institute of Psychology, Sichuan University, Chengdu 610065, China; 3Behavioral Public Administration and Public Leadership Innovation Lab of SPA, Sichuan University, Chengdu 610065, China

**Keywords:** citizen cultivation, equality, autonomy, boundary, respect, normative theory

## Abstract

The concept of “citizen” differs across disciplines. Drawing on insights from multiple disciplines, this paper argues that being respected is the fundamental ethical basis for citizens and proposes a normative framework. In this framework, people being respected as citizens relies on three foundations: autonomy, boundary, and equality. Autonomy means a citizen has the right to make choices and act independently and freely. Nevertheless, this freedom is not boundless. Boundary consciousness is the manifestation of a citizen. Being a citizen implies the presence of boundaries that safeguard individual freedom and autonomy while prohibiting violations of others’ boundaries. Equality is the fundamental element of being a citizen. Subjects in disadvantaged unequal relationships find difficulty in becoming real citizens. People engage in behaviors that harm themselves and others due to differences in social cognition, embedded social relationships that blur behavioral boundaries, and hierarchical cultures that cause status inequality. This theoretical framework can provide ethical and normative guidance for citizen cultivation. Citizen cultivation should focus on fostering people’s autonomy of being responsible and accountable, recognizing and respecting self and others’ necessary boundaries, and constructing individuals’ subjective equality consciousness.

## 1. Introduction

What is citizenship? The “classical civic tradition” was established as early as the ancient Greek period. According to Aristotle, a citizen is an active participant and builder in the public affairs of the polis ([6]), which reflects a proactive civic republican perspective. As history has evolved, the connotations of citizenship have remained in flux. The most influential among them is the theory of citizenship proposed by T.H. Marshall from a sociological perspective, which emphasizes that members of all classes within the nation-state should be equally granted rights and that individual freedom should be safeguarded ([36]). Yet, citizenship is not merely a status granted by the state; it also demands respect and seeks recognition from the state ([52]). Moreover, mutual respect and recognition among citizens are equally important ([39]). In this fluid, globalized society, attributes such as immigration, race, and gender undoubtedly delineate the boundaries of citizenship. Therefore, in the fields of cultural sociology and traditional political theory, some scholars have developed the framework of “cultural citizenship” ([24]), striving for social inclusion and justice for minority and marginalized groups and safeguarding their equal rights ([4]), which exemplify respect for these marginalized populations. Furthermore, the advent of the digital age has given rise to the concept of “digital citizenship”. Although the field of citizenship has shifted, it still emphasizes protecting and respecting the rights and freedoms of every participant who uses the internet effectively while requiring adherence to established systems and ethical standards ([50]).

The definition of citizenship is complex and diverse. However, regardless of the specific connotations or forms, “respect” remains the moral prerequisite and foundation of citizenship. An ideal citizen is an individual who possesses rights and freedoms, and while pursuing their own interests, they respect the rights and freedoms of others and uphold loyalty to the law and regulations ([41]). Therefore, discussions on citizenship should begin with the concept of respect.

“Respect” as the foundation for interpersonal interactions is widely applied in social life and particularly crucial within the context of civic relationships. Only by respecting both oneself and others can one maintain positive interactions within communities and political entities ([19]). However, the exact meaning of “respect” is often unclear to people. True respect means acknowledging the rights and differences of other citizens on an equal basis, constituting a form of mutual recognition among citizens ([46]). The abstract nature of this term makes it easy for people’s understanding of respect to deviate or even be ignored. This may lead to situations where respect is not practiced as expected, resulting in behaviors that undermine civic harmony. For example, when people participate in discussions of public affairs, respect requires listening to the opinions and voices of all citizens, as expressing their views on public events is a right of citizenship and represents the interests of their respective groups. But in fact, many individuals oppose or even attack the opinions of other citizens without careful consideration, depriving others of their rights as citizens and demonstrating a lack of respect for others’ citizenship.

The goal of civic education is to cultivate an ideal and responsible citizen, a process that essentially involves nurturing virtues and moral character. Respect holds value in moral philosophy and serves as the character for virtue cultivation. Being respected has absolute ontological significance. Kant elevated respect for persons to an indisputable absolute significance, arguing that “a person should not merely be valued as a means to achieve the ends of others, or even their own ends, but should be valued as an end in themselves, that is, they possess a dignity (an absolute intrinsic value), which compels all other rational beings to respect them and to regard themselves as equal to any other person on the basis of equality…” ([26]); his notion undoubtedly makes respect the foundation of human virtue cultivation. When people treat others as individuals who possess rationality and agency and who are equal to themselves, they express an attitude of respect towards others. Thus, respect, as a fundamental virtue, should become a consensus among all citizens.

Respect holds not only ontological significance in moral philosophy but also has universal value in secular ethics, serving as a golden rule for handling social relationships. Published in 1948, the Universal Declaration of Human Rights states “All human beings are born free and equal in dignity and rights. They are endowed with reason and conscience and should act towards one another in a spirit of brotherhood”. This statement embodies the general spirit of freedom, equality, and human rights in human civilization, which can be summarized as respect for individuals and the protection of their basic rights and interests. Similarly, all citizens are inherently free and equal, and regardless of their individual characteristics, they deserve full respect ([33]). On this foundation, social relationships can be effectively managed, fostering a democratic and harmonious society.

Within public life, philosophical traditions, and secular ethics, respect serves as a critical regulatory value, making an in-depth analysis of citizenship both theoretically profound and practically relevant. Drawing on a comprehensive review of previous research, this paper claims that respect is the fundamental basis of citizenship. Furthermore, a triple-element theory of citizens from a Chinese pictogram perspective is proposed and explores its normative implications for civic cultivation. Additionally, it examines the underlying individual social cognition, social relationships, and sociocultural factors that contribute to behaviors undermining respect in real-life contexts. These discussions integrate normative considerations of what ought to be with empirical evidence of what is, offering a comprehensive understanding of civic ethics and cultivating awareness of respect among citizens.

## 2. What Is Respect? Dialogues Across Multiple Disciplines

Respect is so important that it has been highly valued in the fields of humanities and social sciences. The notion of respect has long been embedded in the philosophical understanding of human nature. It was not until around the 21st century that respect began to garner widespread attention from many fields, e.g., education, psychology, and ecology. How do scholars across these disciplines define and interpret the concept of “respect”? Does the concept of respect maintain a consistent meaning across disciplines? What intrinsic connections exist between respect and citizenship?

In the realm of philosophers and thinkers, Confucius and Kant are regarded as key figures in defining the concept of “respect”, and many later scholars have cited their viewpoints as evidence. In Confucius’s discourse, “One helps others to take a stand if one desires to take one’s stand, and realizes others if one desires to realize oneself” and “Do not do unto others what you do not want done unto yourself” ([32]) are revered as the “Golden rule” of ethics and interpersonal conduct. In this sense, respect is demonstrated through perspective-taking—a social cognitive process that enables individuals to understand others’ intentions, avoiding actions that others “do not wish for” and fulfilling what others “desire to establish”. This embodies human autonomy—“establish” and “achieve”, that is, the realization of free will; it also implies a boundary awareness, the need to distinguish what is one’s own business and what is others business; at the same time, it means equality, understanding and fulfilling others’ intentions, which cannot be achieved without this premise. In Kant’s “Formula of Humanity”, it is said, “Act in such a way that you treat humanity, whether in your own person or in the person of any other, always at the same time as an end, never merely as a means” ([27]), a principle that encapsulates the essence of respect. First, a person is an end, not a means; thus, the autonomy of a person, or the absolute value of freedom and rationality, is highlighted. Kant believed that “the ability to set oneself a certain purpose is a salient characteristic of humanity (distinguishing it from animality)” ([26]), and to respect oneself and others is to respect the absolute value of this autonomy. Second, recognizing individuals as ends necessitates the establishment of interpersonal boundaries, ensuring that each person can pursue their goals within these limits. Third, Kant’s thought implies that all individuals possess equal dignity ([55]). Only the existence of sharing as the purpose can make people’s freedom and autonomy be realized. Confucius’s “golden rule” and Kant’s “Formula of Humanity” share commonality in the meaning of respect, both adhering to human autonomy, protecting freedom within boundaries, and insisting on equality for all. These principles transcend philosophical theory and become benchmarks for civic behavior. Just as every citizen has the right to freely choose their occupation, provided they assume responsibility for their decisions and adhere to institutional, moral, and legal norms, others have no right to interfere or judge. When citizens uphold these principles, they embody the value of respect.

The function of contemporary school education extends beyond merely sharing information; it also encompasses the moral education of students in their capacity as citizens. ([21]). Since the beginning of the 21st century, a wave of “respect education” has been initiated. The respect education includes the following: “first, respecting the laws of education, fostering students’ upward spirit, teaching students necessary knowledge, and encouraging students’ passion for creativity; second, respecting students as individuals, acknowledging their personality and humanity, learning their interest, and respecting the laws of their physical and mental development; third, fostering students’ awareness of social morals and respect for others” ([45]). Thus, “respect education” is also a form of civic education and aligns with the connotation of cultivating the “Four Citizen” (“四有公民”), who have high ideals, moral integrity, good education, and a strong sense of discipline. Meanwhile, in Malaysian school curricula, respect plays a crucial role in shaping “good citizens” ([46]). All of these reflect the significant importance of respect in the field of education. In respect education, fostering respect for students involves protecting their autonomy, nurturing creativity, and encouraging a love for learning; it is also about setting behavioral boundaries, including adherence to social morals and respect for others. And it lays the foundation for equal dialogue between teachers and students, fostering a sense of equality at the civic level.

Unlike the normative focus of philosophy and education, psychological researchers emphasize the empirical knowledge of “respect”. Empirical research has shown that in real life, respect positively influences interpersonal relationships within organizations, thereby promoting organizational citizenship behavior ([40]). Additionally, citizens’ perception of respect can effectively promote civic engagement ([61]). Studies also indicate a positive correlation between respect, power, and status ([22]). From the perspective of students, respect consists of three dimensions: politeness, dependence–independence, and submission–equality ([10]). The development of students’ attitudes toward respect depends on their ability to engage in independent thinking, social cognition (perspective-taking), empathy, and social interactions with adults and peers. Therefore, as stated, “respect involves regarding oneself and others as independent, free, whole individuals with unique natures, personalities, and dignity” ([34]). Although students may occasionally conflate respect with reverence or admiration, they generally associate respect with concepts such as equal personal status, autonomy, and the maintenance of boundaries and integrity.

When philosophers, educators, and psychologists center their discussions of respect on human beings, ecologists have expanded its scope to include the natural environment. The long-term extensive development of humanity has caused significant environmental issues, prompting reflection on the appropriate relationship between humans and nature. As an essential component of public space, the environment embodies collective interests ([3]). Citizens must recognize that their personal environmental behaviors have the potential to trigger widespread public consequences, as their lifestyle choices affect both other individuals and the natural environment ([25]). Consequently, citizens have the responsibility and obligation to respect and protect it. This means that humans, particularly citizens, should not impose their will upon nature but rather acknowledge the priority, intrinsic value, and natural laws of nature ([63]). Humans, alongside other species, must coexist with nature, and human activities should prioritize the well-being of all living beings rather than serving exclusively human interests. The call of ecological scholars implies several basic assumptions: First, the natural world has autonomy and is not subordinate to human will. This means that the natural world is not merely an object waiting to be developed but an autonomous entity with its own will and inherent laws. Second, humans and nature need to establish boundaries for coexistence, preventing individual activities from harming or interfering with the autonomy of nature and others. Third, humans need to change their mindset, recognizing their place within the natural order. Respecting nature means engaging in equal dialogue with nature and even submitting to the power of nature.

The definition of “respect” varies slightly among academic predecessors due to their different academic backgrounds, especially as ecological scholars extend the object of respect to the natural world, distinguishing their perspective from others. Despite these differences, a common emphasis on equality, autonomy, and boundaries is evident across these discussions. The concept of citizenship inherently incorporates these moral dimensions. Citizens, endowed equally with dignity, must refrain from interfering with the autonomy of others and, in turn, expect their own autonomous decisions to be free from interference, allowing them to pursue objectives that are both morally justifiable and personally meaningful ([14]). Citizens’ respect for others and the environment significantly affects all aspects of public life. Behaviors such as littering or polluting air and water, neglecting environmental conservation, not only damage the natural environment but also violate the rights of other citizens to a clean and healthy environment. Such behavior fails to regard oneself, others, and the environment as equally important, thus undermining the spirit of citizenship and public morality that constitute respect. Hence, it can be observed that respect is the ethical foundation of citizenship.

## 3. The Triadic Theory: Deconstructing the Ethical Foundations of Citizen

From Aristotle’s delineation of citizenship in ancient Greek city-states to the modern citizenship intricately bound to nation-states and ultimately to the concept of global citizenship that transcends cultural and racial divides, citizenship has consistently been closely aligned with the collective. It suggests that citizenship is not to pursue individual desires but to uphold public values. In this context, “private” refers to an individual’s excessive concentration on personal and familial matters, and the concentration is so narrow that it obscures the organic connections between individuals, the state, and society as a whole, ultimately leading to a lack of mutual respect ([47]). Respect, however, serves as the starting point for establishing the fundamental ethics of modern citizenship.

Firstly, while citizens are required to attach importance to collective interests, this does not imply the loss of their independence; they still possess the right to make autonomous choices, including freely expressing their opinions and participating in decision-making processes concerning public affairs. Secondly, this freedom is not unbounded. While individuals retain interests and autonomy in public life, conflicts may arise from divergent interests or viewpoints. Therefore, boundaries must be established for citizens’ behaviors to safeguard the best interests of everyone and the harmony of public life. Lastly, equality is a fundamental requirement for every citizen. Regardless of their social class, race, gender, or other personal characteristics, all citizens are equal. It is precisely based on this equality that the autonomy of each citizen can be freely exercised to a certain degree, thereby minimizing conflicts. In this scenario, respect for oneself, others, and the world is exhibited by every individual, enabling them to truly live as citizens. Consequently, the realization of a “civilized society” or a “world of universal harmony” will increasingly within reach.

In this process, being respected as a citizen comprises three essential elements: autonomy, boundary, and equity, which possess an inherent logical structure as depicted in Figure 1.

### 3.1. Core Element: Granting Autonomy in Choice Is the Core of Respect

Respect, as an attitude that a subject extends towards an object, reflects a scenario where the object of respect has the right to autonomously choose within a specific matter or range. Conversely, losing the right to autonomously choose results in coercion and disrespect. The essence of respect lies in individuals exercising autonomy to access options and make choices according to their will. Autonomous choice is not about the number of options but the freedom of individual will. Even with a single option, an individual retains the freedom to choose or decline. This freedom of will is the key to autonomous choice. Autonomy also serves as the foundation of true human dignity and the source of self-esteem ([12]). Respect is centered on the right to autonomous choice and is integral to civic life. As a political and social ideal, autonomy asserts that individuals should shape their lives according to their own reasons, values, and aspirations. To uphold equal respect for citizens’ autonomy, the nation must treat its citizens neutrally, refraining from favoring any one group for the benefit of certain individuals ([15]). From the perspective of individual citizens, autonomy requires that citizens morally respect themselves and all others possessing behavioral competence ([11]). However, citizens’ autonomy does not emerge spontaneously; rather, it requires support from educators at the political and social levels to develop among individuals ([5]). Therefore, fostering autonomy through civic education is crucial.

Respect, as a reactive relationship, consists of basic elements such as attention, compliance, positive evaluation, and appropriate behavior ([59]). These elements collectively safeguard the object’s right to autonomous choice. When citizens do not receive the attention they deserve, their will is disregarded, and positive evaluations are withheld, leading to a loss of their right to autonomous choice and a sense of disrespect. Scholars have proposed that appropriate behavioral feedback includes actions such as “maintaining distance from him, helping him, praising him or bestowing honor upon him, obeying or following him, not offending or disturbing him, protecting or caring for him, ending him in some way rather than letting him be destroyed in other ways, or speaking of him in a way that reflects his value or status, mourning him, and so on” ([64]). Clearly, these behavioral responses align with an object’s right to autonomous choice, emphasizing the preservation of their autonomous free will.

In summary, safeguarding citizens’ right to autonomous choice is the core manifestation of being respected, with autonomy serving as its core element. When citizens who possess autonomy perceive themselves as respected, they are more likely to actively participate in public affairs. In this process, further exercise autonomy contributes to sustaining a free and democratic societal atmosphere. Most importantly, in a diverse society, autonomy enhances citizens’ understanding and respect for different values and lifestyles, thereby fostering inclusivity and promoting harmonious coexistence among individuals from various backgrounds and beliefs ([44]).

### 3.2. Manifested Element: Boundary Consciousness as the Manifestation of Respect

Citizens’ right to make free and autonomous choices is not unlimited. When autonomy harms or interferes with oneself or others, it must be constrained by boundaries. With society development, these boundaries typically take three distinct forms. The first form is the tangible and visible physical boundary, exemplified by the distinction between private and public spaces. Respect entails a citizen’s freedom to move within their private space and to use and arrange it as they wish. Conversely, the absence or violation of private space also signifies a lack of respect. The second form is the intangible and invisible contractual boundary, established through agreements that define the scope of free choice. For instance, citizens’ behavior is constrained by the law, requiring compliance to prevent crimes and protect the life and property of others. Failure to adhere may lead to severe penalties, including citizenship revocation. The third form is the meta-boundary, designed to protect the autonomy of people’s choices. As the saying goes, “Freedom should not come at the cost of giving up freedom”. If individuals are permitted to abandon their right to free and autonomous choice, they risk being treated as mere means rather than ends. For instance, in democratic countries, selling one’s political vote, under the pretense of “free choice”, is actually a violation of the right to free political elections of citizenship, which is illegal. The meta-boundary is established to maintain human autonomy itself. Essentially, this is also a manifestation of people’s autonomy, redefining the scope and degree of self-respect to establish meta-boundaries that protect the rights and interests of free autonomy.

It is evident that beyond the core of free and autonomous choice rights, there exist tangible or intangible and visible or invisible boundaries that ensure the realization of people’s free and autonomous rights within a limited scope. Boundaries are necessary because societal resources and conditions supporting autonomous choice rights are finite. Therefore, boundaries need to be established to ensure that all parties can exercise their free and autonomous choice rights within a certain scope. The necessity of boundaries also lies in the fact that humans cannot abandon their free will and perish, which is a necessary constraint for human nature to be respected. Without boundaries, human freedom and autonomy may lead to greed, selfishness, and unrestrained indulgence, ultimately undermining these very principles.

### 3.3. Fundamental Element: Equality as the Foundation of Respect

The full realization of citizens’ autonomy requires that all citizens hold equal status. The first article of the 1948 Universal Declaration of Human Rights states, “All human beings are born free and equal in dignity and rights. They are endowed with reason and conscience and should act towards one another in a spirit of brotherhood”. Without this equal status, citizens’ autonomy cannot be effectively realized, and the boundaries of free will may also be eroded. For example, in feudal societies, respect was hierarchical rather than mutual, often emphasizing non-reciprocal and non-universal respect, such as “inferior to superior”, “young to old”, “student to teacher”, and “female to male” ([58]). Such respect preserved the free choice rights of those in dominant positions while sacrificing, to some extent, the free and autonomous choice rights of those in subordinate positions. The concept of “citizenship” signifies the repudiation of traditional hierarchical structures and emphasizes the institutionalization of equality in modern societies ([54]). Therefore, the development of modern civilization has institutionally established the principle of “equality for all”, laying the foundation for individuals to receive the respect they deserve.

To respect others is to treat them with equality ([35]), acknowledging their status and identity as equal members of a moral society ([13]). However, social development inevitably results in objective inequalities among citizens. This is particularly evident in contemporary electoral practices, where voting tendencies favor more privileged social classes, including the elderly, males, the highly educated, and the affluent, while the interests of vulnerable groups may be insufficiently protected ([16]). Thus, the right to respect is consequently threatened. To bridge the gap between ideal aspirations and reality, human society has developed two basic forms of equality: institutional equality and subjective equality.

Institutional equality refers to the establishment of the equal status of people through systemic and institutional means, such as the provisions for the equal status of all in the Universal Declaration of Human Rights and the constitutions of various countries. For example, to promote gender equality, many countries and regions have increased the number and selection of female candidates in political positions through legislation or party deliberations ([31]). In some African authoritarian states, women also have opportunities to enter cabinets and take up positions ([30]). In terms of economic development, countries like Denmark and Sweden implement fiscal policies such as high taxation to sustain their welfare programs ([9]). This approach not only mitigates inequality arising from the market economy but also fosters economic dynamism. By contrast, the US federal government provides unemployment benefits, retraining allowances, and support for relocation, job searching, and healthcare through Trade Adjustment Assistance (TAA) for citizens who lose their jobs due to global economic competition ([2]).

Subjective equality, on the other hand, refers to fostering the belief that “everyone is equal” in citizens’ subjective perceptions. The real world inevitably faces horizontal social comparisons that create distinctions between those in advantageous and disadvantageous social positions. If these comparisons and distinctions cause disadvantaged citizens to perceive inequality subjectively, social trust may deteriorate ([29]), potentially leading to demands for benefit redistribution and driving policy reform ([7]; [20]). Psychologically instilling the belief that everyone is equal is crucial to ensuring that citizens’ rights, especially the free and autonomous rights of those in weaker social positions, which often depend on whether those in advantageous social positions have established the value of “everyone is equal”. Only when citizens in advantageous positions, such as politicians and the wealthy, internalize the value of treating others as equals with equal dignity and rights and collectively participate in public life ([18]) can genuine democracy and equality be promoted.

Institutional equality and subjective equality are often mutually structured. On the one hand, institutional equality lays the foundation for citizens to form the concept of equality for all in their psychology and subjective value concepts. Because of the equal regulations in the social system, citizens can consciously maintain the equal status of all in their psychology and behavior. On the other hand, subjective equality, as the connotation of civilized concepts, has been proposed and promoted as an important value of intellectual enlightenment during the transition of hierarchical societies, which also laid the ideological conditions for writing “equality for all” into social systems. In the actual operation of society, various social phenomena of inequality are also constrained by institutional discipline, curbing the inflated free and autonomous rights of those in advantageous positions and safeguarding the basic social welfare guarantees for those in disadvantaged positions. Therefore, equal status is the cornerstone of being respected.

In summary, in the triadic theory structure of the ethical foundations of citizens, the right to free and autonomous choice is the core meaning of being respected. To realize citizens’ free and autonomous choice rights, it is necessary to establish a corresponding sense of boundaries, all of which are based on equal status. Once equal status is lost, the rights to free and autonomous choice and the corresponding boundaries cannot be guaranteed, and respect becomes impossible to talk about. Even with the premise of equal status, once the right to free and autonomous choice is hindered, it is a sign of disrespect. Such behavior often occurs through the act of overstepping boundaries. Establishing equal status, setting necessary boundaries, and exercising free and autonomous rights within the boundaries are the ways to realize the attitude of respect.

## 4. Normative Guidance of the Triadic Theory of Respect for Citizen

The triadic theory of being respected as a citizen is synthesized and refined based on the perspectives of predecessors from various disciplines, especially normative viewpoints, and empirical observations. Therefore, it also provides certain normative guidance for managing various civic relationships in social life. This paper analyzes the ethical guidance brought by the triadic theory of respect in four aspects: the relationship between the subject self and object self, the relationship with others, the relationship within groups, and the relationships between groups.

### 4.1. Handling the Relationship Between Subject Self and Object Self

Self-respect, which refers to respecting oneself, arises from the interactions between civic individuals (the subject) and their own selves (the object). Previous scholars have often viewed self-esteem as a subjective need for individuals to have their behaviors and abilities recognized or validated by others and society ([62]). In Maslow’s hierarchy of needs, this is positioned at the fourth level, representing a higher-level need. Within the framework of the triadic theory of respect, self-esteem means that, in the judgment of the civic subject, the autonomy of the self is in a state of potential realization, possessing the rights and opportunities for free and autonomous choice.

In Rawls’s view, self-respect is a primary good, and therefore, the design of social systems should support and enhance the self-respect of citizens ([56]). For instance, when the government grants citizens freedoms of speech, thought, and political rights, ensuring that citizens always perceive that they have the room and flexibility to make choices, citizens’ self-esteem is fostered. However, excessively high self-esteem can evolve into a desire for an overly expansive space for choice, leading to a narcissistic sense of superiority that threatens the interests of other citizens. To appropriately maintain self-esteem, it is necessary to have corresponding boundaries that align with the right to free and autonomous choice. On one hand, these boundaries typically exist to ensure that citizens do not interfere with the free and autonomous choice rights of others, allowing for a broad space that is self-sufficient without hindering others. On the other hand, these boundaries also often serve to protect the civic subject’s own rights to free autonomy. If the boundaries are too narrow, they may hinder the exercise of an individual citizen’s autonomy, thereby lowering self-esteem; if they are too wide, they may infringe upon the autonomy of others, leading to inflated self-esteem and potential narcissism.

Self-esteem also entails recognizing oneself as having equal rights and status as others in the context of being a citizen. It is understood that when an individual possesses the same rights as others, those others are endowed with equivalent rights as well. ([43]). When citizens perceive and acknowledge themselves as being in a disadvantaged position in interpersonal comparisons, their self-esteem may suffer, potentially leading to self-destructive behaviors, such as self-harm or suicide. Conversely, if citizens perceive and acknowledge themselves as being in an advantageous position, their self-esteem may become excessively inflated, resulting in behaviors that harm other citizens, such as aggression ([28]). In real life, the objects of horizontal social comparisons among citizens are often selectable, leading to what can be described as an elastic state of self-esteem, where individuals may feel “less than” when comparing themselves to those above them and “more than” when comparing themselves to those below them. When the perceived inequalities in status do not match the actual inequalities, individuals may experience blind overconfidence (perceiving an advantage while actually being at a disadvantage) or anxious self-doubt (perceiving a disadvantage while actually being at an advantage). In the realm of inevitable socioeconomic disparities among citizens, for example, citizens of lower socioeconomic status, when comparing themselves to one of higher status, showing lower levels of self-respect ([42]), may develop skepticism about their own living conditions and even foster dissatisfaction with society. Alternatively, a citizen who exhibits blind confidence in their socioeconomic status may hold contempt for those of lower status.

### 4.2. Handing Relationships Between Civic Individuals and Others

Mutual respect occurs in the domain of relationship management between civic individuals and others. Starting from the triadic theory of citizen being respect, mutual respect implies that both parties’ rights to free and autonomous choice can be realized, that there are certain boundaries to protect the free and autonomous rights of both parties, and that both parties have an equal status.

The realization of the right to free and autonomous choice is about promoting the autonomy, creativity, and positive growth of the other as a human being. Ideally, both parties act in ways that are most conducive to the realization of each other’s autonomous choice rights, acknowledging, praising, and admiring each other’s behavior. In real life, conflicts often arise between the free and autonomous choice rights of both parties. For instance, in the communal spaces of Chinese communities, elderly individuals possess the right to enjoy public life by engaging in square dancing. But when dancing occurs at inappropriate times and with excessively loud music, it inevitably disrupts the daily lives of other residents. Both parties must reach a consensus through negotiation. Yet, these consultations require institutional support to proceed smoothly. Communities, as key entities in national social governance, play a vital role in promoting citizen dialogue. They may enact noise control rules and designate specific activity zones to balance the rights of the elderly with the needs of other residents. In such scenarios, the interacting parties can try their best to provide options that satisfy both.

However, when there is only one possibility, one party’s free will must be compromised. The parties then need to communicate their free will to reach a compromise and agreement, adjusting their boundaries to adapt to the new interactive scenario. This adjustment process often has a temporal and situational flexibility. Temporally, it satisfies one party at one time and another at another time; contextually, it satisfies one party and the other in a different situation. This interactive flexibility is also a kind of psychological contractual boundary, maintaining the realization of the free and autonomous choice rights of both interacting parties.

Communication and coordination between interacting parties are predicated on an equal status. If there is no equal status between the interacting parties, it can evolve into the tyranny of the advantaged over the disadvantaged. If the subject occupies a dominant position, they might completely disregard the object’s free will. Due to the patriarchal nature of traditional society, men have enjoyed a certain dominant position over women, leading to the development of “male chauvinism” and the bad custom of disrespecting women ([17]). In the distribution of opportunities pertaining to civic participation, numerous “ordinary” young people are excluded, whereas those who are deemed to possess the potential to emerge as future leaders are advantaged ([57]). Of course, in discussing equality as the basis for citizens’ respect, it has been noted that many countries reduce these inequalities through policies and legislation, striving to ensure that all citizens have equal opportunities for dialogue and consultation regardless of gender or social background. As can be seen, the tripartite theory of respect can explain this phenomenon and also offer direction for normative efforts.

### 4.3. Handing Relationships Between Civic Individuals and Groups/Organizations

The mutual respect between civic individuals and groups/organizations is based on “contracts”, referring to the formal and informal contractual relationships established between individuals and groups or organizations. Bound by contracts, a hierarchical relationship is formed, and the mutual respect between individuals and organizations can be explained within the framework of the triadic theory of respect.

When a citizen respects a group or organization, it means that on contractual matters, the group or organization is in a state of free and autonomous choice. This requires the citizen to be loyal to their duties, completing the agreed-upon tasks with the group on time, in quantity, and with quality. In this way, the group remains in a state of having choices. Should the citizen neglect their duties, the group would face difficulties in that specific matter, necessitating the activation of emergency and punitive measures. Conversely, the group’s respect for the citizen means that the group should match the citizen’s rights according to the contract and create conditions to promote the individual’s autonomy. It is more likely for citizens to demonstrate behaviors that are beneficial to the organization, thereby enhancing organizational performance ([38]). If the group exploits the citizen, arbitrarily reduces the benefits they are entitled to, or fails to grant sufficient autonomy in the execution of their work, the citizen may choose to leave or resist, leading to conflicts, ruptures, and termination of the relationship.

The formation of this contractual relationship often stipulates the behavioral boundaries and the scope of free and autonomous rights in the form of written or oral agreements. The contract itself implies the boundaries of both parties’ behavior, which often need to be presented in a visual, inspectable, and supervisory manner, such as contracts or agreements. If a civic individual or group fails to complete the agreed-upon matters, they should be subject to organizational punishment that matches their dereliction of duty. The establishment of these boundaries allows civic individuals and groups to clarify their responsibilities, rights, and interests to clearly understand where their behavioral boundaries lie and to ensure the freedom of action within those boundaries. And the most powerful form of this contract is none other than the national legal system. Legal frameworks, such as labor laws, consumer protection laws, and constitutional principles, aim to clarify the rights, responsibilities, and boundaries of action for both contractual organizations and individual citizens.

In addition, this kind of contractual relationship, where responsibilities, rights, and interests are equivalent, must be based on the premise of equality and voluntariness from both parties. Without this premise, the contractual relationship may not be reached, or it may lead to despotism and exploitation of the individual by the group or organization. As the guarantor of contractual fairness, the state relies on modern legal systems to prevent exploitation and oppression, ensuring that no party is forced into an agreement that violates their dignity or fundamental freedoms. If this premise is lost, such contractual relationships are generally considered void under the law.

In analogy to the relationship between citizens and the state, citizens are afforded rights by the state, while they are concurrently obligated to fulfill their duties and adhere to legal boundaries; failure to do so results in legal sanctions. Similarly, state institutions at all levels are mandated to uphold citizens’ individual rights in accordance with the law; otherwise, citizens may engage in resistance against the state in pursuit of their autonomous rights ([8]). Therefore, the relationship between civic individuals and groups/organizations still needs to be based on equality and centered on promoting the realization of both parties’ rights to free and autonomous choice and agreeing on acceptable behavioral boundaries (i.e., contract content). In this way, both parties to the contract can receive the respect they are entitled to.

### 4.4. Handing of Relationships Between Groups

The mutual respect between groups is similar to the mutual respect between civic individuals and can also be explained within the framework of the triadic theory of respect. Mutual respect between groups also means that each group can assert its own internal affairs, negotiate, adjust and establish group boundaries, and have an equal status in getting along in the sense of a group.

Within a country, there are various subcultural groups. Each group has its own customs and traditions. In the case of no interference between groups, each group should equally enjoy the right to develop freely and independently. In multireligious countries, where many citizens hold diverse religious beliefs, there is a necessity for mutual respect and tolerance among different religious groups, despite their differing faiths ([60]).This kind of respect is different from the respect in the sense of individual interaction. The object of respect is often not a specific person but institutions, conventions, customs, cultures, traditions, etc., based on groups. Some scholars also call this institutional respect ([23]). In constructing institutional respect, the state utilizes the education system and public discourse to help individuals understand the values of different religions and cultures ([37]). For instance, in some predominantly Islamic countries, legal protections for religious freedom are in place to promote religious tolerance and mutual respect ([49]).

Mutual respect between groups is also centrally reflected in the way countries get along with each other, as nations are essentially aggregations of citizens and representatives of their collective will. In 1954, the Agreement on Trade and Communications between China’s Tibet Region and India proposed “mutual respect for sovereignty and territorial integrity, mutual non-aggression, non-interference in each other’s internal affairs, equality and mutual benefit, and peaceful coexistence”. These five principles have gradually become the basic norms for handling international relations. In 2023, Xi Jinping delivered a keynote speech titled “Walking Hand in Hand on the Path to Modernization” at the High-level Dialogue between the Communist Party of China and World Political Parties. For the first time, he proposed the “Global Civilization Initiative”, of which the first item is “jointly advocate respecting the diversity of world civilizations”. He further elaborated that “humans only differ in skin color and language, and civilizations only differ in their colorful varieties, but there is absolutely no distinction of high or low, superior or inferior.”; “Shoes need not be the same; what matters is that they fit”.

The above political propositions are in line with the triadic theory of respect. When each group or country has the right to make autonomous decisions on its internal affairs without interference from other groups or countries, this group or country, along with its citizens, is respected. “Non-interference in each other’s internal affairs” means respecting the autonomy of each country at the national level and ensuring the realization of the right of each country to make choices freely and independently. Therefore, in international diplomatic work, “non-interference in each other’s internal affairs” is an important principle for political consultations.

However, being respected at the group and national levels is not easy. History mercilessly records that “if you are backward, you will be beaten”. Therefore, the proposition of equality among countries in politics has its political civilization significance and is also the premise for mutual respect among countries. When equality cannot be established between groups and countries, there is a high probability of power politics by those in dominant positions, and even conflicts and wars. Recognizing the equal status of various groups and countries between groups and countries beyond the disparities in actual development has modern civilization significance.

## 5. The Mechanism of Deviation from “Respect” and Ethical Solutions

Based on an analysis of previous scholars’ understanding of respect and citizenship, this paper constructs a normative theory to deconstruct the theoretical components and structure of the ethical foundations of citizens—the triadic theory of respect. This theory provides guidance in managing relationships among civic individuals and groups. The relationship between civic individuals and groups often evolves into the relationship between group representatives or leaders and group members. Inter-group relationships are also frequently realized through the relationships between group representatives. Therefore, the relationships among civic individuals, civic groups, and between them can all be referenced by the management of relationships between individuals. So, based on the triadic theory of respect, what factors might lead a citizen to deviate from an attitude of “respect”?

### 5.1. The Impact of Empathy on Attitudes of Respect: The Social Cognition Process of Autonomous Intentions

Starting from the triadic theory of being respected as a citizen, a civic subject deviating from an attitude of respect implies a disregard for, devaluation of, and interference with the free and autonomous rights and interests of other subjects.

If a civic subject fails to understand the autonomous intentions of another subject, it affects the cultivation of a respectful attitude. The most important social cognitive mechanism in this context is perspective-taking, which is the capacity of a subject to view issues from the object’s point of view. If a subject cannot adopt the object’s perspective, they cannot fully understand the intentions of others, thereby hindering their free and autonomous rights and leading to a lack of respect. For instance, in cross-cultural communication, if a civic subject cannot comprehend certain cultural taboos from the other party’s perspective, it may result in offending the other party and causing feelings of disrespect. Such situations are typically not intentional but can create barriers in communication.

If a civic subject can understand the autonomous intentions of an object but chooses to act against them, it then evolves into a conscious act of disrespect, such as intentionally embarrassing someone. In this case, although the subject has some perspective-taking ability, they may lack the social cognitive mechanism of empathic concern, failing to grasp the emotional state of the other subject when they are treated with disrespect or to transfer this emotional state to themselves. A deficiency in either perspective-taking ability (cognitive empathy) or empathic concern (affective empathy) may lead a subject to reject the autonomous intentions of others, resulting in behavior that disrespects others.

### 5.2. The Interplay of Social Relationships and the Deconstruction of Respect: The Social Dissolution of Boundaries

Boundaries have sociocultural attributes. In independent individualistic cultures, civic individual boundaries are relatively clear, and people place a greater emphasis on their rights to free and autonomous choice, whereas in interdependent collectivist cultures, civic individual boundaries are more blurred, and the free and autonomous rights of individuals are often influenced by significant others with whom they are connected. This interplay of social relationships often has a deconstructive effect on attitudes of respect.

Due to the ambiguity of boundaries, civic individuals may interfere in the private affairs of significant others. For example, in Chinese family relationships, it is common for members of the family of origin to offer guidance on the life arrangements of members of the secondary family, which is often seen as wise counsel, with the adage warning, “Disregard the elderly’s advice, and suffer the consequences immediately”. If members of the secondary family adhere to traditional views and accept such guidance, they may not feel disrespected. However, with the changes in sociocultural values and an increased emphasis on individual autonomy, members of secondary family often resist such arrangements and perceive this so-called guidance as a serious breach of boundaries.

### 5.3. Hierarchical Culture’s Threat to Attitudes of Respect: The Process of Social Restructuring for Equality

Equality is a prerequisite for respect; if an object does not possess a status equal to that of the subject, it is difficult to ensure their rights to be respected. Social hierarchy is a fact, and there are often distinctions of status in many aspects. Research evidence indicates that socioeconomic status has a significant positive effect on self-esteem ([53]), with higher socioeconomic status leading to higher levels of self-esteem. In both Western bureaucratic structures, characterized by their hierarchical systems, and traditional Chinese Confucian thought, which emphasizes the order based on seniority and the distinction between the close and distant, the existence of a hierarchical culture is consistently reflected. Under this cultural backdrop, whether citizens can gain respect often depends on whether those with superior social status are willing to treat them as equals, thereby preserving their rights to freedom and autonomy. This hierarchical cultural background undermines the psychological foundation of equality among all, thus threatening attitudes of respect. The development of modern civilization aims to eliminate various forms of social inequality and restore the dignity of being human to every individual. Therefore, the systems of nations and United Nations declarations alike include the principle that all people are born equal in their charters, making efforts to reconstruct psychological and social equality. Actual social reality has not yet eliminated inequality, which means that a significant number of people have not been granted the respect and dignity they deserve ([48]). It is for this reason that the triadic theory of respect also provides normative guidance for a dignified and respectful public ethical life.

In education and public ethics, creating a social and institutional environment where everyone is equal, and maximizing the realization of individual freedom and autonomy without hindering or harming others’ boundaries, is the greatest protection and promotion of the dignity of the educated and citizens. This means that in the implementation of education, there is a need to vigorously develop equal education, boundary education, and positive education. Equal education strives to establish the concept of equality among all individuals as citizens, boundary education creates conditions for individuals to identify and follow the autonomous boundaries of themselves and others, and positive education provides a safeguard for maximizing individual autonomy, tapping into human potential, and realizing value. These three types of education are integral to achieving the goals of civic education. Civic education advances democratic values and enhances social cohesion by cultivating citizens with a sense of responsibility, respect, and inclusivity, laying the foundation for national development ([51]). Thus, governmental efforts are necessary to integrate those into educational systems through policy-making and to provide the necessary resources and support to ensure its implementation and optimize educational quality ([1]).

## 6. Conclusions: Autonomy, Boundary, and Equality as the Triadic Structure of Being Respected as a Citizen

While the triadic theory of respect can provide a framework for understanding attitudes of respect in various social relationships, it is important to note the scope of its application. This theory is primarily aimed at social interaction processes within human society and offers limited guidance for the management of human–nature relationships. This is largely because the relationship between humans and nature may be unequal, and humans need to revere nature rather than simply respect it. It requires a deep understanding of natural laws and compliance with these laws to achieve human well-being. Therefore, this theory does not have appropriate explanatory power for understanding human-nature relationships due to this inequality. Similarly, this theory is not applicable to individuals, groups, and societies that recognize hierarchical social structures and can only be used to understand and interpret social relationships that develop on the basis of equality.

In summary, the triadic theory of respect provides new knowledge for understanding the three conditions and their interrelationships necessary for an object to be respected in human social interactions. An object’s respect is inseparable from the protection of its rights to free and autonomous choice, from the specific boundaries that provide the necessary space for self-comfort without hindering others, and from the equal status that serves as the fundamental guarantee. This also guides our efforts towards a society and era where everyone is respected and everyone has dignity.

## Figures and Tables

**Figure 1 behavsci-15-00513-f001:**
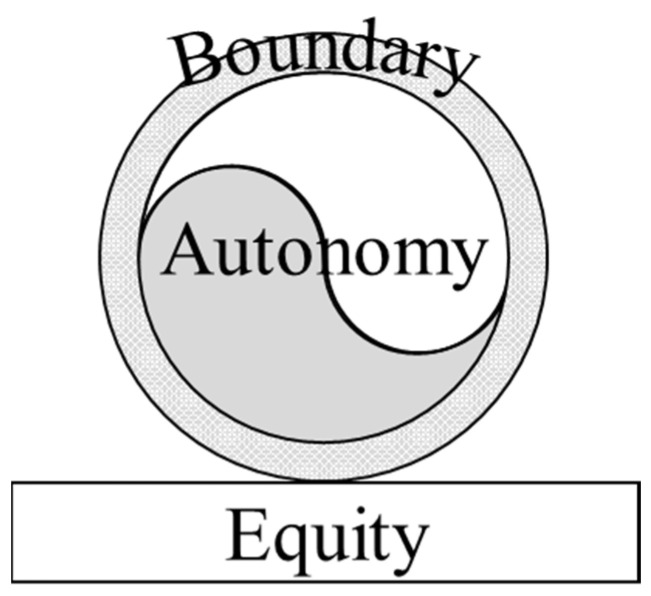
The interrelationship of the components in the triadic theory of respect.

## Data Availability

No new data were created or analyzed in this study.

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
