# Peer review of "The Ethical Foundations of Being Respected as a Citizen: A Triadic Theory Consisting of Autonomy, Boundary, and Equality"

_behavsci, 2025, doi:10.3390/bs15040513_

Round 1
Reviewer 1 Report
Comments and Suggestions for Authors
This paper examines the concept of "citizen" across multiple disciplines and argues that respect is the fundamental ethical basis of citizenship. It proposes a framework built on three foundations: autonomy, boundary, and equality. Autonomy grants individuals the freedom to act independently, but this freedom is limited by necessary boundaries that prevent harm and uphold mutual respect. The framework suggests that true citizenship requires fostering autonomy with responsibility, recognizing personal and social boundaries, and promoting equality to counteract social hierarchies and power imbalances.
The article addresses an interesting and important topic with potential for further development. However, the presentation at times appears somewhat inconsistent, which may make it challenging for the reader to follow the line of argumentation. The argumentation could benefit from further elaboration to enhance analytical depth, and the theoretical level of precision could be improved to meet the standards of an academic journal. The structure and presentation give the impression of a textbook chapter rather than a research article, and a clearer academic grounding could strengthen the article’s contribution to the field.
I also believe the article overlooks a crucial element: the role of the state in establishing a foundation for its citizens to be respected. The state plays a vital role in upholding individual rights, protecting freedoms, and fostering a society built on dignity and equality. This responsibility extends across several key areas directly related to the article’s discussion. The state must ensure access to essential services and social security, enabling individuals to live with dignity. It should implement economic policies that create opportunities for all and reduce inequality. Additionally, the state has a duty to protect individuals from violence, exploitation, and abuse - whether by other citizens, organizations, or even state actors. By establishing and enforcing laws that safeguard fundamental rights like freedom of speech, religion, and assembly, the state lays the groundwork for a just society. Without addressing these aspects, the article’s analysis of the ethical foundation of respect in citizenship remains incomplete and lacks depth.
Author Response
Reviewer 1
Comment #1: This paper examines the concept of "citizen" across multiple disciplines and argues that respect is the fundamental ethical basis of citizenship. It proposes a framework built on three foundations: autonomy, boundary, and equality. Autonomy grants individuals the freedom to act independently, but this freedom is limited by necessary boundaries that prevent harm and uphold mutual respect. The framework suggests that true citizenship requires fostering autonomy with responsibility, recognizing personal and social boundaries, and promoting equality to counteract social hierarchies and power imbalances.
Response: Thank you very much for well summarizing our main points. We benefited a lot from your following suggestions and further improved the quality of our paper.
Comment #2: The article addresses an interesting and important topic with potential for further development. However, the presentation at times appears somewhat inconsistent, which may make it challenging for the reader to follow the line of argumentation. The argumentation could benefit from further elaboration to enhance analytical depth, and the theoretical level of precision could be improved to meet the standards of an academic journal. The structure and presentation give the impression of a textbook chapter rather than a research article, and a clearer academic grounding could strengthen the article’s contribution to the field.
Response: Thank you for your comment. Given that you don’t directly point out where we have inconsistent presentations, we have thoroughly reviewed the whole text and tried our best to identify whether there are some inconsistencies. We found that the inconsistencies might be due the term use of ‘citizen’ and ‘citizenship’. “Citizen” is used to denote individual citizens, while “citizenship” refers to the rights, obligations, or an abstract state of citizens. We have now revised some of the term based on the connotations so as to convey the accurate argumentation. Please refer to the tracked changes. As to the analytical depth and theoretical level of precision, we also tried our best to reframe some of the points, ensuring that the theoretical construction level is linked to the actual phenomenal observations and the basic philosophy behind. We hope our revisions have resolved your concern.
Comment #3: I also believe the article overlooks a crucial element: the role of the state in establishing a foundation for its citizens to be respected. The state plays a vital role in upholding individual rights, protecting freedoms, and fostering a society built on dignity and equality. This responsibility extends across several key areas directly related to the article’s discussion. The state must ensure access to essential services and social security, enabling individuals to live with dignity. It should implement economic policies that create opportunities for all and reduce inequality. Additionally, the state has a duty to protect individuals from violence, exploitation, and abuse - whether by other citizens, organizations, or even state actors. By establishing and enforcing laws that safeguard fundamental rights like freedom of speech, religion, and assembly, the state lays the groundwork for a just society. Without addressing these aspects, the article’s analysis of the ethical foundation of respect in citizenship remains incomplete and lacks depth.
Response: Thank you for your insightful suggestions. However, we would like to kindly draw your attention to the content in the sections where we have discussed your concern, e.g., ‘Manifested Element: Boundary Consciousness as the Manifestation of Respect’ at line 308-313 of page 7. We also acknowledge that the original manuscript lacked sufficient discussion on this aspect. To address this concern, the following changes have been made in the revised manuscript:
- In “3.3 Fundamental Element: Equality as the Foundation of Respect”, we emphasize the state's role in promoting women's political participation and addressing economic inequality (at line 361 to 374 of page 8).
- In “4. Normative Guidance of the Triadic Theory of Respect for Citizen”, we highlight the government's importance in safeguarding citizens' freedoms of speech, thought, and political participation (at line 428 to 432 of page 9).
- We discuss the role of communities as government agents in facilitating citizen deliberation (at line 471 to 484 of page 10).
- We examine how the state ensures contractual relationships between citizens, organizations, and itself through legislation (at line 526 to 556 of page 11).
- We further clarify the state's role in promoting mutual respect among multicultural and multireligious groups (at line 563 to 575 of page 12).
- Finally, we underscore the government's crucial role in developing civic education (at line 691 to 697 of page 16).
Reviewer 2 Report
Comments and Suggestions for Authors
I really liked the article and I appreciated the arguments developed to improve the debate on the concept of citizenship through the philosophical concept of respect with reference to Immanuel Kant's thought. My question to the author concerns the following issue: for Kant, respect is a fundamental moral obligation; respect is owed to all individuals on the basis of being rational beings and is not dependent on social standing, achievements, or personal character - the concept of equality that the author introduces as well. However, the author recognizes the existence of hierarchical structures and -consequently- of conflicts. There the author suggests that the human society has formed two basic forms of equality: institutional equality and psychological equality. There the author would need maybe to be more precise about institutional equality that cannot be reduced to positive actions in University, raising how the issue of the conflict between equality measures imposed by state and economic competition for example in free market theories. As for the psychological equality, this is also problematic, as we pass from the ethical and the political sphere to another level.
"Social systems should be designed to support and enhance citizens' self-esteem, as it is crucial for individuals to effectively develop and realize their moral capabilities (Whit- 401 field,2017).Good self-esteem implies that citizens consistently feels they have the space and flexibility to make choices." There is a sort of confusion between self-esteem, moral capabilities and freedom of choices and opportunities.
These are just some of the aspects that can be further developed, as more attention to the political participation dimension in respect to the concept of citizenship.
Comments on the Quality of English LanguageJust some orthograph problems (but I am anot English mother language)
Author Response
Reviewer 2
Comment #1: I really liked the article and I appreciated the arguments developed to improve the debate on the concept of citizenship through the philosophical concept of respect with reference to Immanuel Kant's thought. My question to the author concerns the following issue: for Kant, respect is a fundamental moral obligation; respect is owed to all individuals on the basis of being rational beings and is not dependent on social standing, achievements, or personal character - the concept of equality that the author introduces as well. However, the author recognizes the existence of hierarchical structures and -consequently- of conflicts. There the author suggests that the human society has formed two basic forms of equality: institutional equality and psychological equality. There the author would need maybe to be more precise about institutional equality that cannot be reduced to positive actions in University, raising how the issue of the conflict between equality measures imposed by state and economic competition for example in free market theories. As for the psychological equality, this is also problematic, as we pass from the ethical and the political sphere to another level.
Response: We appreciate the reviewer’s concern regarding institutional equality and psychological equality. To further elaborate on institutional equality, we have revised the original discussion: we emphasize the state's role in promoting women's political participation and addressing inequalities stemming from free market economies and global economic competition (at line 361 to 374 of page 8). However, regarding psychological equality, what we actually intended to convey is the idea of "subjective equality." This concept is indispensable because absolute equality is unattainable in real life. If citizens cannot establish subjective equality at the psychological level and embrace the value of equality for all, they will be unable to confront inequalities in reality. A lack of perceived subjective equality among disadvantaged citizens may affect social harmony, while citizens in advantageous positions might rely on their superiority and further overlook the challenge of disadvantaged groups. Further clarification has been provided in the revised manuscript (at line 375 to 395 of page 8).
Comment #2: "Social systems should be designed to support and enhance citizens' self-esteem, as it is crucial for individuals to effectively develop and realize their moral capabilities (Whit- 401 field,2017). Good self-esteem implies that citizens consistently feels they have the space and flexibility to make choices." There is a sort of confusion between self-esteem, moral capabilities and freedom of choices and opportunities.
Response: Thank you for your comment. We have checked the cited articles and made revisions, rephrasing the relationship between self-esteem and freedom of choice. Further clarification has been provided in the revised manuscript (at line 428 to 432 of page 9). “In Rawls' view, self-respect is a primary good, and therefore, the design of social systems should support and enhance the self-respect of citizens (Whitfield, 2017). For instance, when the government grants citizens freedoms of speech, thought, and political rights, ensuring that citizens always perceive that they have the room and flexibility to make choices, thereby fostering citizen’s self-esteem.”
Comment #3: These are just some of the aspects that can be further developed, as more attention to the political participation dimension in respect to the concept of citizenship.
Response: Thank you very much for your suggestion. We have further elaborated some parts of the arguments and also included some discussions about political participation in the paper.
Comment (about quality of English language): Just some orthograph problems (but I am anot English mother language)
Response: Thank you for your comment. We sincerely apologize for any spelling or grammatical errors. We have proofread the article once again to ensure the use of correct English expressions.
Round 2
Reviewer 1 Report
Comments and Suggestions for Authors
The authors have done a good job revising the article. I believe that the article can be published in its present form.
Author Response
Thank you for your time and effort.